# Similarities and Disparities of e-Commerce in the European Union in the Post-Pandemic Period

Rodica Manuela Gogonea [1], Liviu Cătălin Moraru [2] , Dumitru Alexandru Bodislav [2] , Loredana Maria Păunescu [3] and Carmen Florentina Vlăsceanu [4],*

1    Department of Statistics and Econometrics, The Bucharest University of Economic Studies,
     010374 Bucharest, Romania; manuela.gogonea@csie.ase.ro
2    Faculty of Theoretical and Applied Economics, The Bucharest University of Economic Studies,
     010374 Bucharest, Romania; liviu.moraru@economie.ase.ro (L.C.M.);
     dumitru.bodislav@economie.ase.ro (D.A.B.)
3    Department of Cybernetics, Economic Informatics, Finance and Accounting, Petroleum-Gas University,
     100520 Ploiesti, Romania; loredana.paunescu@upg-ploiesti.ro
4    Doctoral School of Business Administration, The Bucharest University of Economic Studies,
     010374 Bucharest, Romania
*    Correspondence: carmen_vlasceanu@yahoo.com

**Abstract:** The emergence of the COVID-19 pandemic has resulted in notable transformations of the commerce landscape, particularly in the realm of electronic commerce. This sector has experienced a precipitous advancement, characterized by substantial modifications of online business undertakings, encompassing both products and services. The aim of the current research was to explore the similarities and differences between European Union member states in the context of e-commerce in the post-pandemic period, taking into consideration the population's level of education, the risk of poverty, as well as households' access to the internet. The analysis was conducted for the year 2021, which represented the most recent year for which data were available, and was based on the application of the hierarchical cluster methodology, which included the Ward method and the Robust Tests of Equality of Means (Welch and Brown–Forsythe). Five clusters resulted, which included a minimum of three countries and a maximum of nine. The present study focused on examining the similarities and disparities within clusters, as well as among countries belonging to those clusters. These observed similarities and disparities are believed to be the outcome of various indicators that influence the realm of electronic commerce, and they are contingent upon the economic development level of each country and their ability to cope with the challenges posed by the COVID-19 pandemic. The information obtained in this study pertains to the future of electronic commerce in the sense of identifying premises that allow the development and application of development strategies.

**Keywords:** electronic commerce; clusters; similarities and disparities; customer relationship management (CRM) and secure transactions; risk of poverty; level of internet access of households; education level of the population; COVID-19 pandemic

## 1. Introduction

The COVID-19 pandemic has had a significant impact on consumer markets and their preferences, leading to substantial changes. This has resulted in the heightened priority of trade in goods and services conducted through electronic platforms. The current economic recovery hinges upon the ongoing process of digitization, wherein e-commerce serves as a vital driver of economic activity. The increasing utilization of electronic commerce for both domestic and international sales necessitates a corresponding diversification of products to cater to a more diverse consumer demand structure [1,2]. The rise of the COVID-19 pandemic has led to an escalation in competition within the e-commerce sector. The process of enticing consumers presents a complex and nuanced challenge, necessitating

the development of tailored strategies that align with the distinct cultural traditions and consumption behaviors exhibited within specific countries [3,4].

The purpose of this research is to elucidate the differences and similarities among the member states of the European Union in the aftermath of the COVID-19 pandemic, particularly with regard to the influence of educational attainment, poverty vulnerability, and household internet accessibility on electronic commerce. The e-commerce analysis entails the examination of key factors such as the involvement of economic agents in e-commerce, customer relationship management (CRM), and the security of transactions.

The Digital Economy and Society Index (DESI) 2022 [5] underscores the significance of digital competencies in fostering economic advancement and mitigating the potential for digital marginalization. This is particularly relevant as essential services increasingly migrate to online platforms, thus encouraging potential negative repercussions and impacting the overall quality of life. The present context also encompasses an analysis of the indicator pertaining to the level of internet access in households.

This study highlights the significance of considering the risk of poverty and the level of education as crucial factors for analysis. The degree of economic development has a substantial impact on the risk of poverty, exhibiting an inverse relationship with e-commerce. Consequently, heightened levels of economic development correspond to diminished levels of poverty and present obstacles to the growth of e-commerce. In relation to the classification of variables that focus on the population's educational attainment, it is noteworthy that the predominant proportion of the overall population should pertain to tertiary education levels 5–8, with a secondary emphasis on individuals with higher secondary education and post-secondary, non-tertiary education (levels 3 and 4), as they are poised to potentially drive future e-commerce activities.

The current research conducted by the Digital Economy and Society Index (DESI) identifies strategic objectives aimed at attaining the Digital Decade targets of the European Union. The objectives seek to attain a population proficient in digital literacy, a labor force equipped with advanced digital competencies, a robust and sustainable digital infrastructure, the promotion of digitization within the commercial sphere and the modernization of public services through digital methodologies.

Following this, the European Union (EU) has implemented measures to enhance levels of digital literacy. Enhancing academic standards is crucial to attaining this objective, given that an individual's digital proficiency is significantly impacted by their educational attainment. This assertion aligns with the results outlined in the Digital Economy and Society Index (DESI). The primary emphasis is placed on the utilization of indicators to elucidate the extent of education, thereby rendering a more nuanced and comprehensive evaluation in contrast to data that have been aggregated. Additionally, a comprehensive analysis of the population is undertaken to ascertain its impact as a present and future catalyst for the advancement and expansion of e-commerce initiatives.

The study addresses a notable research gap by examining the impact of five selected variables on e-commerce activity in twenty-seven European countries. The investigation of the present status of e-commerce growth, through an analysis of the interconnected variables and their ramifications for the e-commerce sector post-COVID-19 in 2021, presents a research deficiency necessitating the formulation and execution of comprehensive strategies within this domain.

Simultaneously, we endeavor to elucidate the manner in which modifications of variables can be understood in light of their classification by country subsequent to the onset of the COVID-19 pandemic.

The clustering of countries accentuates their similarities, as each cluster comprises nations that exhibit similar characteristics across all six indicators. Disparities between nations persist and are perpetuated through comparative analyses within each grouping.

The article is comprised of five distinct sections. Following the introductory section, the subsequent section (Section 2) provides a comprehensive review of the specialized literature concerning electronic commerce. This literature encompasses various perspectives on

electronic commerce as well as the key variables involved in its analysis, as discussed by experts in the field and other individuals with a vested interest in the subject. Section 3 provides an in-depth explanation of the variables, data collection method, and other essential elements that are required for the successful implementation of the clustering methodology. This section lays the groundwork for the application of the clustering method. The dataset comprises six key variables reflecting the socio-economic conditions in the year 2021, following the global pandemic. These variables include: (1) e-commerce, customer relationship management (CRM), and secure transactions—PECO; (2) poverty risk rate—PPR; (3) level of internet access in households—PHIA; (4) population with lower educational attainment (levels 0–2)—PPEL02; (5) population with upper secondary and non-tertiary post-secondary education (levels 3 and 4)—PPEL34; and (6) population with tertiary education (levels 5–8)—PPEL58. The subsequent sections, namely Sections 4–6, will showcase the findings, succeeded by a thorough discussion and the ultimate conclusions. The findings pertaining to the variables are analyzed in conjunction with the ranking of the clusters and the countries within each cluster. The focus of the analysis is primarily on the average maximum and minimum percentages, as well as the extremes observed at the country level, to elucidate the positions within the ranking and the range of variation. The analysis of the aforementioned findings elucidates both the favorable and adverse effects exerted by the indicators on e-commerce, as well as their magnitude following the onset of the COVID-19 pandemic in 2021.

The current study adds to the existing body of literature on e-commerce activity by providing complementary insights and findings. The objective of this article is to contribute to the advancement of research on electronic commerce through an updated study, taking into consideration the aftermath of the COVID-19 pandemic. This study aims to provide a foundation for the development and implementation of strategies to enhance this commercial domain.

## 2. Literature Review

The rapid transformations of e-commerce that occurred over time determined the gradual appearance of many structural changes in its composition. But, starting in 2019, the COVID-19 pandemic accelerated this evolution exponentially, changing the way of life, as most activities had to be carried out online. In this context, the importance of using digitalization on a large scale has increased, with electronic commerce expanding and developing rapidly in all countries of the world, including in European countries.

Electronic commerce (e-commerce), it can be stated, represents the act of sale and purchase, in close interconnection with their related activities (negotiation, transmission of information, etc.), carried out through an electronic platform on the internet [6–14].

E-commerce, like any field of activity, has advantages for both companies and consumers but also presents some disadvantages compared to the traditional commerce. For the consumer, the main advantages would be the non-stop accessibility of the market, because a product can be returned if it does not meet expectations (even over a longer period of time), as well as high transparency. In addition, some of the sites show the opinions of other buyers about the product, facilitating additional information about the respective product. Saving time remains the main advantage of e-commerce, considering the fact that for the procurement of products or service it is no longer necessary to walk to registered commercial units. But e-commerce also has some disadvantages for the consumer: one cannot try certain products or touch them before buying them, certain goods cannot be chosen and individually picked (e.g., fruits, vegetables, etc.). However, considering all these disadvantages, mainly due to saving time, it remains a popular form of products purchasing, especially after the COVID-19 pandemic [14].

As far as firms are concerned, e-commerce facilitates international exchanges and these may increase their sales, even if some barriers to these exchanges may appear [15,16]. The main advantage consists of increasing the number of customers and implicitly sales, because the companies are now able to sell even outside their country. Globally, the

majority of the largest companies by turnover are those that conduct predominantly online commerce [17]. It should be mentioned here that within the EU, there is a single market, which facilitates the free movement of goods, services, capital and people. As a result of these provisions, there are no customs duties for the delivery of products within the European Union, this being an advantage for the companies in the member states but also for consumers who have the ability to purchase cheaper products, without paying customs duties from other EU states [18]. Also, e-commerce can reduce transaction costs and make communication between customers and firms more efficient while increasing the speed of transactions [14]. A further limitation arises from the influence of unfavorable online reviews on consumer decision-making, resulting in swift changes in consumer preferences for products or companies [19].

The main disadvantages for the companies identified by the authors of the article are the barriers that can appear in electronic commerce, among which it can be mentioned: problems related to internet access, qualified personnel, problems related to distribution—they do not have their own fleet and no companies to whom they can collaborate. At the same time, since some of the goods cannot be tried/tested physically by the consumer before purchase, there will be a higher return rate, which will increase the company's costs (particularly those related to transport, packaging, etc.), which constitutes another significant disadvantage.

The evolutionary trends observed in the domain of e-commerce in recent years can be attributed to the impact of the global COVID-19 pandemic. The transmission and manifestation of COVID-19 caused a significant amount of fatalities and had a major adverse effect on the global economy. This impact was particularly pronounced in stock markets, leading to adverse consequences [19–23].

Gradually, the entire world economy has been affected, starting with China, and in the state of global crisis e-commerce took on an important role, which continues to gradually expand, gaining larger dimensions. The effects were different depending on the nature of the goods (food or non-food) and the method of trade (traditional trade or online trade) [24,25]. Drastic measures, even lockdowns, which prohibited people from leaving their homes without a well-founded reason, had the effect of transforming the supply process from a direct one into an on online based orders, through which most of the goods reached consumers. As a result, even though the general trade declined worldwide, the e-commerce has greatly increased [10,17,26].

E-commerce is becoming the "lifeline" for many companies that had to close their physical stores. The impact of the pandemic on the online commerce manifested itself differently depending on different economic sectors. Travel ticket sales were down (as expected due to COVID-19 restrictions), but homewear sales were up. The introduction of this system had positive effects on courier companies and click-and-collect delivery services [27] in specific urban areas, leading to a significant increase in the number of parcels delivered, sometimes doubling the previous amount [28]. Customers have become much more attentive to courier services and have begun to evaluate them in terms of their quality and punctuality [29]. A negative correlation between age and online shopping was noted, with young people having a greater predisposition to online shopping than older people [30]. In this context, the consumer experience and trust in online transactions has increased and they have started to face a rapid evolution [31–33].

The favorable encounters consumers had with online shopping platforms amidst the pandemic prompted them to persist in this behavior even subsequent to the relaxation of restrictions [34]. E-commerce is recognized as a sector that has exhibited sustained high levels of activity even in the post-COVID-19 pandemic period, characterized by an ongoing process of substantial growth. In contrast to other industries, wherein online enterprises experienced considerable surges during the pandemic, their subsequent progression following the pandemic did not exhibit comparable momentum. One illustrative instance involves the emergence of online communication platforms such as ZOOM and Google Meet during the pandemic. These platforms, while initially developed as a response to the

prevailing circumstances, have shown a recent inclination toward a resurgence in personal, in-person interaction, as compared to electronic commerce [35].

As humanity faced the COVID-19 pandemic, many specialists paid special attention to electronic commerce, studying the phenomenon more intensively, the course of the processes and its effects. Some of the effects were gradually measured and used in other research studies in order to construct an overall picture, or to analyze in more detail aspects regarding the development of e-commerce. In this regard, one of the significant indicators used in such research can be mentioned: e-commerce, customer relationship management (CRM) and secure transactions. The data for this indicator are collected annually by the National Statistical Institutes (INSSE) based on a questionnaire developed by Eurostat [36]. The model of this questionnaire changes annually in order to correctly measure the use of new technologies and at the same time not to burden the respondents, meaning certain questions are asked every 2 or even 3 years and therefore the time series may have interruptions. The questionnaire is addressed to enterprises with 10 or more employees that have online sales (e-commerce), excluding the financial sector. For 2022, a sample of 151,000 companies was selected from the 1.47 million companies with at least 10 employees. The results are expressed in the form of weights, representing the number of enterprises that present a certain feature, weighted by the total number of enterprises. The importance of transaction security for e-commerce is obvious, because the main issue of e-commerce refers to maintaining the integrity of transactions, confidentiality and correctness of data, considering that these security measures are intended to limit the negative effects both in the online environment and outside of it [37,38].

The implementation of security measures for variables within the cluster is additionally justified by the importance attributed to this domain by the European Union, as demonstrated by the incorporation of "Safety" as a distinct category in the Digital Competence Framework 2.0, which constitutes a revision of the DSI methodology.

The population's hesitance to engage in online transactions stems from a perceived lack of confidence in the security of personal data and monetary transfers, as acknowledged in the previous literature [39–42]. However, it is important to note that these studies do not encompass EU countries, and they do not definitively demonstrate the strong correlation between them through the utilization of the hierarchical cluster methodology.

Our research seeks to address this deficiency and contribute to the existing knowledge. The evolution of e-commerce is closely related to the level of economic development of each country, which can be transposed through the risk of poverty. The at-risk-of-poverty rate indicator is calculated as a weight and tells us the percentage of the total population living in households where the disposable income is below 60% of the median disposable income per adult-equivalent at the national level in the current year and also within 2 years from the previous 3 years. It is determined based on various data collected by the European Union Statistics on Income and Living Conditions (EU-SILC) over a period of at least 4 years according to the methodology [43].

This correlation has not previously been investigated; thus, our study endeavors to address this gap in the existing literature. One of the key repercussions of poverty for online commerce is the inherent impediment posed by limited financial resources, resulting in restrained purchasing power. In some studies, poverty is considered a "disease" that needs to be treated in order to benefit from future economic development [44]. In addition to presenting an impediment, inadequate financial resources pose a challenge when seeking to access the internet, as it necessitates possession of a computer and the means to procure a paid internet subscription. In various long-term scholarly investigations, empirical evidence suggests that children hailing from disadvantaged backgrounds tend to exhibit diminished academic achievement when compared to their counterparts from affluent households [45]. This observed disparity in educational performance subsequently engenders reduced financial prospects, thereby establishing a self-perpetuating cycle wherein poverty is sustained. Several studies suggest that e-commerce has the potential to alleviate poverty, particularly in areas with limited access to technology, such as rural

environments [46]. The aforementioned correlation can also be substantiated by research findings suggesting that in Africa, which is characterized by prevalent poverty, the advent of mobile phones alongside diminished communication costs has fostered enhanced efficacy within the labor market and overall welfare [47].

The examination of aspects concerning the population is indispensable when discussing the evolution of e-commerce, as it serves as the principal driving force in this sphere of enterprise. The level of education of the population is pertinent in regard to the perception of this online phenomenon, as well as the adaptability of internet users. Education plays a pivotal role in fostering digital commerce by instructing subsequent cohorts to discover, analyze, and employ relevant information, ultimately cultivating their capacity to comprehend economic–social and existential occurrences. Moreover, education imparts essential skills in harnessing computer technology for practical endeavors, encompassing online transactions and commercial endeavors.

Education is widely regarded as a significant determinant of consumer behavior, particularly in the realm of e-commerce, as reflected by a multitude of scholarly works. This influence is derived from factors such as digital proficiency, technology familiarity, and foreign language proficiency [48]. Simultaneously, consumer education holds considerable significance in the prioritization of needs, enabling individuals to establish a resilience toward forceful promotional tactics online while discerning genuine necessities from artificially generated desires [49]. The provision of consumer education to individuals starting from the primary cycle holds significant importance [50].

The selection of the educational attainment level as a representative indicator for e-commerce is justified by the given context. This statement presents data regarding the proportion of the overall population that has accomplished a specific educational milestone, classified according to the International Standard Classification of Education (ISCED) coding system for educational attainment. In the assessment of educational attainment, the primary criterion considered is the successful completion of the most advanced level of education with official recognition in the form of a qualification or diploma. However, in cases where lower levels of education do not provide such credentials, the acquisition of admission to a higher level is deemed sufficient evidence of graduation. Furthermore, it is noteworthy that both vocational and general education hold significance in the academic discourse [51].

Other research studies focused on analyzing the "Internet access level for households" as an indicator in order to correctly measure the use of new IT technologies and monitor the development of e-commerce. Internet access is essential for the ability to surf the internet for multiple purposes, either for online shopping or accessing various sources of educational information or advice related [52].

Several studies indicate that consumer ownership of electronic devices and the quality of internet services play significant roles in determining the success of e-commerce. Thus, the degree of economic development appears to have a lesser impact on online sales [53–55].

The findings indicate that the COVID-19 pandemic has had a lasting and beneficial impact on digital trade activity. This impact arises from a significant diversification of the consumer and product categories within the digital environment. Moreover, this diversification is occurring amidst an overarching commitment to sustainability across all economies worldwide. The evolution of e-commerce is also based on the strategies developed and applied at the local, regional or national level in relation to the level of development of each country [14,56–58].

For this statistical–econometric research, the accuracy of the results obtained is indicated by the confidence level of 95% used to verify (test) the research hypotheses:

**H1.** *The risk of poverty significantly influences, inversely proportionally, the level of use of electronic commerce at the level of all countries.*

**H2.** *The level of internet access of households significantly and directly influences the level of use of e-commerce.*

**H3.** *The level of use of e-commerce depends directly and significantly on the level of education of the population at the level of all countries.*

### 3. Data Series and Methodology

In order to identify and highlight the disparities and similarities between European states regarding the impact of education or poverty on the level of e-commerce, the research methodology included the following stages:

Identification and collection of variables included in the research.

Analysis of their characteristics through the lens of the application conditions of the hierarchical cluster methodology.

Effective application of the hierarchical cluster methodology, which involves:

- ❖ Proximity matrix generation
- ❖ Generation of clusters
- ❖ Testing the statistical significance of the estimated averages of the variables at the level of the generated clusters

Analysis of the obtained results.

The current investigation commenced by outlining its research aims, with a specific emphasis on the identification and analysis of variations and similarities among European Union member states pertaining to the relationship between education levels and poverty risk, as well as the correlation between internet access and transaction security. Following this, six discrete datasets were selected for the purpose of analysis, as delineated in Table 1. Several factors contribute to the phenomenon, including economic entities with e-commerce capabilities, household internet access levels, poverty risk, and educational attainment levels within the population.

**Table 1.** The list of variables included in the analysis.

| Variables | Significations of Variables | UM |
|---|---|---|
| PECO | E-commerce, customer relation management (CRM) and secure transactions | % |
| PPR | At-risk-of-poverty rate | % |
| PHIA | Level of internet access—households | % |
| PPEL02 | Population with less than primary, primary and lower secondary education (levels 0–2) | % |
| PPEL34 | Population with upper secondary and post-secondary non-tertiary education (levels 3 and 4) | % |
| PPEL58 | Population with tertiary education (levels 5–8) | % |

Source: elaborated by the authors.

The selection of variables was informed by the DESI study and the variables identified within, with the intention of proposing practical resolutions to the identified issues. In order to achieve proficiency in digital literacy, it is imperative that the populace receives appropriate education. The rationale for selecting the three stages of education lies in the expectation that a higher level of education will result in greater internet usage and consumer activity compared to individuals with lower levels of education. In the context of the Digital Economy and Society Index (DESI), we contend that household internet access serves as a reliable proxy for sustainable digital infrastructure, while the poverty-risk rate effectively represents poverty vulnerability, thereby potentially contributing to negative effects on overall quality of life. The article emphasized the investigation into the potential additional and interpenetrative relationships between the selected variables and electronic commerce. This investigation suggests that such relationships should not be overlooked in policy considerations.

The data were obtained from the Eurostat database [59]. The cluster analysis described in this study was conducted as a cross-sectional assessment of the rules pertaining to a specific point in time. The data series utilized in this analysis pertained to the most recent year for which data were accessible, specifically the year 2021.

Table 2 displays the principal features of the six variables. Upon conducting the individual analysis, it can be inferred that the distributions exhibit platykurtic tendencies and a relatively symmetrical disposition, characterized by variation amplitudes ranging from 15.65% to 29.40%. In regard to the level of homogeneity exhibited within the data series, it is noted that only the PHIA series demonstrates homogeneity. The PPEL34 and PPEL58 series are observed to be relatively homogeneous, while the remaining series display heterogeneity, indicating notable disparities between EU member states with respect to the values of the variables PECO, PPEL02 and PPR.

**Table 2.** The main characteristics of the variables included in the analysis.

| Parameters | PECO | PPEL02 | PPEL34 | PPEL58 | PPR | PHIA |
|---|---|---|---|---|---|---|
| Mean | 24.29 | 21.55 | 46.27 | 32.19 | 14.36 | 91.85 |
| Standard Error | 1.56 | 1.55 | 2.00 | 1.48 | 0.86 | 0.78 |
| Median | 22.60 | 20.10 | 46.40 | 34.20 | 12.50 | 91.88 |
| Standard Deviation | 8.09 | 8.04 | 10.38 | 7.68 | 4.46 | 4.06 |
| Sample Variance | 65.44 | 64.72 | 107.76 | 59.04 | 19.92 | 16.46 |
| Kurtosis | −0.82 | 0.75 | −0.69 | −0.57 | −0.91 | −0.43 |
| Skewness | 0.31 | 1.11 | 0.02 | −0.29 | 0.48 | −0.13 |
| Range | 28.40 | 29.40 | 38.90 | 28.80 | 16.00 | 15.65 |
| Minimum | 11.80 | 10.90 | 25.60 | 16.40 | 7.70 | 83.53 |
| Maximum | 40.20 | 40.30 | 64.50 | 45.20 | 23.70 | 99.18 |
| Variation Coefficient | 0.33 | 0.37 | 0.22 | 0.24 | 0.31 | 0.04 |
| Confidence Level (95.0%) | 3.20 | 3.18 | 4.11 | 3.04 | 1.77 | 1.60 |

Source: calculated by authors using SPSS.

Considering these factors, along with the recognition that the comprehensive examination of the six variables can yield noteworthy supplementary insights into the positioning of the European Union member states, the hierarchical cluster methodology was selected as the preferred method for grouping. For this, the matrix Z was generated:

$$Z = \left\| z_{ij} \right\|_{i=\overline{1,n}, j=\overline{1,m}} \tag{1}$$

In (1), n is the number of states subject to the analysis (n = 27) and m is the number of variables taken into account (m = 6). The proximity matrix was generated starting from the Z matrix and employing the square of the Euclidean distance as a metric [60]:

$$W = \left\| w_{ij} \right\|_{i=\overline{1,n}, j=\overline{1,m}}, \ w_{ij} = \sum_{i=1}^{n} \left( z_{ik} - z_{ij} \right)^{2}, \ j = \overline{1,m}, \ k = \overline{1,m} \ j \neq i, \ k \neq i, \ w_{ii} = 0 \tag{2}$$

To determine the distance between clusters, Ward's method was used. There being two clusters, A and B, and $x_i$, an item to include in a cluster, the distance between A and B is defined as follows:

$$\Delta(A, B) = \sum_{i \in A \cup B} \| x_i - m_{A \cup B} \|^2 - \sum_{i \in A} \| x_i - m_A \|^2 - \sum_{i \in B} \| x_i - m_B \|^2 - \frac{n_{A \cap B}}{n_{A \cup B}} \| m_A - m_B \|^2 \tag{3}$$

In (3), $m_i$ is the centroid and $n_i$ is the number of elements from clusters i.

The Robust Tests of Equality of Means (Welch and Brown–Forsythe) were used to test the statistical significance of the cluster means. The hypotheses are:

$H_0$: there is no significant difference between the means of variables at the cluster level.

$$\exists \ m_i = m_j, \ i = \overline{1,r}, \ j = \overline{1,r}, \ i \neq j \tag{4}$$

$H_1$: there is a significant difference between the means of the analyzed variables.

$$m_i \neq m_j, \ \forall \ i = \overline{1,r}, \ j = \overline{1,r}, \ i \neq j \qquad (5)$$

The condition for accepting the null hypothesis ($H_0$) was:

$$F_{stat} < F_{\alpha, df_1, df_2} \text{ equivalent to Sig.F} > \alpha \qquad (6)$$

The processing and the obtained results assumed a confidence coefficient of 95% ($\alpha = 0.05$).

The main used tools were SPSS and Excel with the Real Statistic Resource Pack [61].

### 4. Results

Through a thorough examination of the six variables and subsequent statistical tests and significance analyses, the present EU member states were categorized into five distinct groups, as shown in Table 3. Cluster C has the highest numerical value in terms of its cluster size, whereas cluster E presents the lowest number of constituent states.

**Table 3.** Cluster structure.

| Cluster | Structure |
|---|---|
| A | Belgium, Denmark, Ireland, Netherlands, Sweden |
| B | Bulgaria, Czech, Croatia, Poland, Romania, Slovakia |
| C | Germany, Estonia, Greece, Latvia, Lithuania, Hungary, Austria, Slovenia, Finland |
| D | Spain, Italy, Malta, Portugal |
| E | France, Cyprus, Luxembourg |

Source: elaborated by authors using SPSS.

The cluster generation dendrogram is illustrated in Figure 1.

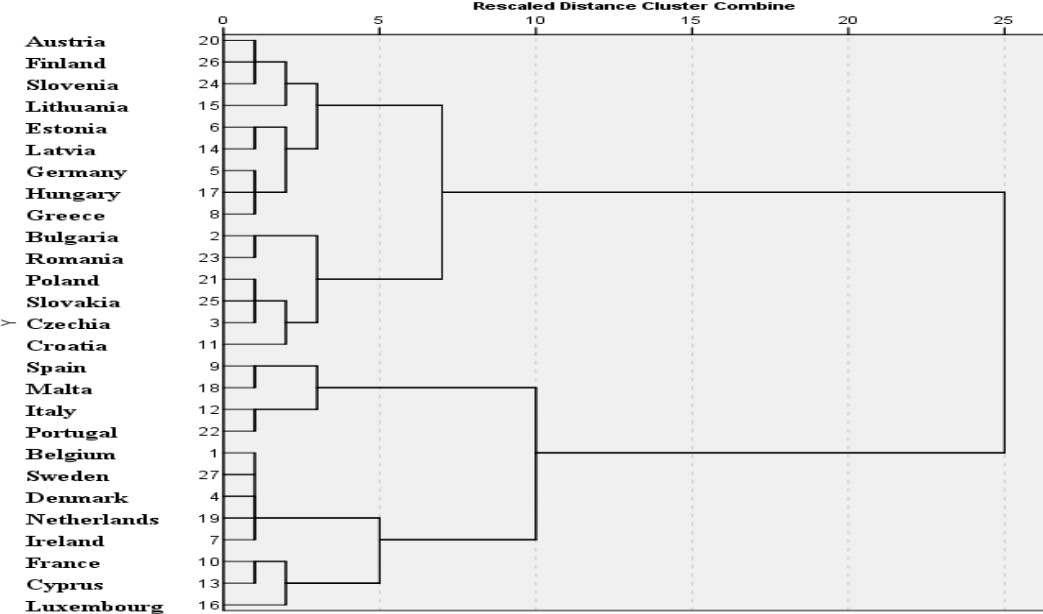

**Figure 1.** Cluster generation dendrogram. Source: own elaboration based on Eurostat data.

Based on the comprehensive analysis of the six variables and subsequent tests and significance analyses, the present member states of the European Union were categorized into five clusters (Table 3). Cluster C has the highest count among all the clusters, whereas cluster E exhibits the lowest number of states.

This current research on the similarities and discrepancies in e-commerce involves an analysis of the means of the variables, primarily among the clusters, with a focus on the ranking of the clusters in relation to the percentage means of the variable PECO (e-commerce, customer relationship management (CRM), and secure transactions). The features presented serve to further substantiate the statistical significance of the mean values, as confirmed by the Welch and Brown–Forsythe tests.

The analysis of the results (see Table 4) entails primarily a comparison of the average percentage values of the indicators across the five clusters, with particular attention paid to the ranking of the clusters. This analysis can be conducted with consideration of the following aspects:

➢ In the ranking of the clusters, the low weights of the PPR poverty risk (fourth, fifth places) reflect the high level of PECO electronic commerce (first, second places), which highlights the validation of hypothesis I1.

➢ The validation of hypothesis I2 is demonstrated by the countries at the cluster level that exhibit a high level of development, representing a significant level of internet access among PHIA households (reflected in high weights placing the countries in the first and second positions), which determines a high level of PECO electronic commerce (places one, two).

➢ The educational attainment of the PPEL population is a critical and discernible factor in determining their utilization of e-commerce, irrespective of their specific educational level (PPEL02, PPEL38, PPEL58). The PPEL02 fund is identified as having low significance for e-commerce, while the PPEL38 and PPEL58 funds are currently active and notable indicators in the field. Their high weights in the rankings (first and second places) have a substantial impact on the level of e-commerce, customer relationship management, and secure PECO transactions, thus providing insight into why the countries in the analyzed clusters occupy the top two positions. The findings indicate that the observed direct correlations among these variables provide support for the validation of hypothesis I3.

➢ The third place serves as an intermediary position, where each variable exerts a varying degree of influence on e-commerce in correlation with the interplay and specificities of the country or cluster in question.

**Table 4.** Robust tests of equality of means.

| | | Statistic [a] | df$_1$ | df$_2$ | Sig. |
|---|---|---|---|---|---|
| PECO | Welch | 7.685 | 4 | 8.733 | 0.006 |
| | Brown–Forsythe | 7.012 | 4 | 18.031 | 0.001 |
| PPEL02 | Welch | 30.359 | 4 | 8.773 | 0.000 |
| | Brown–Forsythe | 30.831 | 4 | 20.054 | 0.000 |
| PPEL34 | Welch | 43.718 | 4 | 7.442 | 0.000 |
| | Brown–Forsythe | 25.348 | 4 | 7.653 | 0.000 |
| PPEL58 | Welch | 11.160 | 4 | 8.052 | 0.002 |
| | Brown–Forsythe | 9.571 | 4 | 10.412 | 0.002 |
| PPR | Welch | 16.349 | 4 | 9.163 | 0.000 |
| | Brown–Forsythe | 3.299 | 4 | 12.915 | 0.045 |
| PHIA | Welch | 4.092 | 4 | 8.209 | 0.041 |
| | Brown–Forsythe | 4.236 | 4 | 15.668 | 0.016 |

[a] Asymptotically F distributed. Source: elaborated by the authors using SPSS.

Further examination of the similarities and disparities within clusters incorporates an assessment of the defining variables, considering them in terms of statistical indicators such as the mean, standard deviation and standard error. In addition, confidence intervals,

determined by the minimum and maximum percentages, are used to estimate the mean by the lower and upper limits (95% conf. int. for the mean).

The analysis of the characteristics of the clusters, as presented in Tables 5 and 6, serves as a basis for evaluating and comparing the average values of the analyzed variables (PECO, PPEL02, PPEL34, PPEL58, PPR, PHIA). These means, particularly bounded by their 95% confidence intervals, establish reference points for performing ranking and comparative analyzes as part of the research study.

**Table 5.** The main characteristics of the six variables analyzed for the first three clusters ranked in relation to e-commerce, customer relationship management (CRM), secure transactions (PECO).

| Cluster | Variable | Mean | Std. Dev. | Std. Err. | 95% Conf. Int. for Mean | | Min | Max |
|---------|----------|------|-----------|-----------|--------|--------|-----|-----|
| | | | | | Lower | Upper | | |
| C_A | PECO | 34.88 | 5.07 | 2.27 | 28.58 | 41.18 | 28.00 | 40.20 |
| | PPR | 11.28 | 1.30 | 0.58 | 9.66 | 12.90 | 9.50 | 12.40 |
| | PHIA | 95.55 | 2.70 | 1.21 | 92.19 | 98.90 | 92.30 | 98.56 |
| | PPEL02 | 21.88 | 3.00 | 1.34 | 18.16 | 25.60 | 17.70 | 25.30 |
| | PPEL34 | 38.70 | 1.32 | 0.59 | 37.06 | 40.34 | 37.10 | 40.10 |
| | PPEL58 | 39.40 | 3.80 | 1.70 | 34.68 | 44.12 | 34.90 | 45.20 |
| C_C | PECO | 24.97 | 6.01 | 2.00 | 20.35 | 29.59 | 17.10 | 36.00 |
| | PPR | 15.51 | 4.99 | 1.66 | 11.68 | 19.35 | 9.70 | 23.70 |
| | PHIA | 91.32 | 3.65 | 1.22 | 88.51 | 94.12 | 85.07 | 96.56 |
| | PPEL02 | 17.39 | 4.08 | 1.36 | 14.25 | 20.53 | 10.90 | 23.50 |
| | PPEL34 | 49.73 | 2.90 | 0.97 | 47.50 | 51.96 | 46.30 | 55.40 |
| | PPEL58 | 32.91 | 4.49 | 1.50 | 29.46 | 36.36 | 25.40 | 39.80 |
| C_D | PECO | 23.15 | 6.28 | 3.14 | 13.16 | 33.14 | 17.10 | 29.20 |
| | PPR | 16.68 | 0.93 | 0.47 | 15.19 | 18.16 | 15.90 | 17.90 |
| | PHIA | 91.07 | 3.56 | 1.78 | 85.41 | 96.74 | 87.34 | 95.92 |
| | PPEL02 | 37.85 | 2.75 | 1.37 | 33.48 | 42.22 | 34.00 | 40.30 |
| | PPEL34 | 34.13 | 7.38 | 3.69 | 22.39 | 45.86 | 25.60 | 42.90 |
| | PPEL58 | 27.98 | 7.70 | 3.85 | 15.72 | 40.23 | 17.80 | 36.50 |

Source: elaborated by authors using SPSS.

Table 5 shows the characteristics of clusters C_A, C_C and C_D, which occupy the first three places in terms of the PECO variable (34.88% cluster C_A, 24.97% cluster C_C and 23.15% cluster C_D). Following the conclusion of the COVID-19 pandemic, three distinct clusters have emerged, comprising nations with the highest degree of e-commerce development. The present analysis examines the clusters with the objective of elucidating the extent of the other variables and their corresponding associations with PECO.

The findings from the initial cluster, comprising Belgium, Denmark, Ireland, The Netherlands, and Sweden (C_A), indicate that these countries exhibit the most advanced e-commerce capabilities, customer relationship management (CRM), and secure transaction practices, holding the leading position in the ranking with 38.88%. The diminished prevalence of the poverty risk in these countries, as indicated by the 11.28% rate (ranking fourth among the clusters), provides evidence supporting hypothesis I1, which posits an inverse relationship between this indicator and electronic commerce. The analysis underscores the substantial impact of advanced economic development on e-commerce, as evidenced by the top ranking (95.55%) of these nations in terms of the household internet access levels (PHI), thereby confirming hypothesis I2. At the cluster level, the fulfillment of hypothesis I3, which posits that the extent of electronic commerce utilization is directly

and considerably influenced by the population's level of education, is evidenced by the significant positions of two (PPEL_58) and three (PPEL_02 and PPEL_34) out of the three education-representing variables. The analysis undertaken considers the primary nations that proficiently adjusted to the changes following the COVID-19 pandemic, promptly formulating and implementing e-commerce strategies tailored to the economic circumstances of each respective country.

**Table 6.** The main characteristics of the six variables analyzed for the last two clusters ranked in relation to e-commerce, customer relationship management (CRM), secure transactions (PECO).

| Cluster | Variable | Mean | Std. Dev. | Std. Err. | 95% Conf. Int. for Mean | | Min | Max |
|---------|----------|------|-----------|-----------|-------|-------|------|------|
| | | | | | Lower | Upper | | |
| C_B | PECO | 19.15 | 7.04 | 2.87 | 11.76 | 26.54 | 11.80 | 29.70 |
| | PPR | 15.78 | 5.61 | 2.29 | 9.90 | 21.67 | 7.70 | 21.60 |
| | PHIA | 88.34 | 3.12 | 1.27 | 85.07 | 91.61 | 83.53 | 92.42 |
| | PPEL02 | 16.27 | 4.19 | 1.71 | 11.87 | 20.66 | 12.00 | 21.60 |
| | PPEL34 | 60.20 | 4.00 | 1.63 | 56.01 | 64.39 | 53.30 | 64.50 |
| | PPEL58 | 23.55 | 4.29 | 1.75 | 19.05 | 28.05 | 16.40 | 29.10 |
| C_E | PECO | 16.43 | 3.84 | 2.22 | 6.89 | 25.98 | 12.00 | 18.80 |
| | PPR | 10.07 | 1.40 | 0.81 | 6.59 | 13.55 | 8.50 | 11.20 |
| | PHIA | 95.31 | 3.35 | 1.94 | 86.97 | 103.64 | 93.33 | 99.18 |
| | PPEL02 | 22.30 | 2.71 | 1.56 | 15.58 | 29.02 | 19.70 | 25.10 |
| | PPEL34 | 36.80 | 5.77 | 3.33 | 22.47 | 51.13 | 30.40 | 41.60 |
| | PPEL58 | 40.90 | 4.19 | 2.42 | 30.49 | 51.31 | 36.30 | 44.50 |

Source: elaborated by authors using SPSS.

The analysis will further expound upon the correlation between the decline in the ranking of the PECO indicator (comprising e-commerce, customer relationship management (CRM), and secure transactions) and its implications. Specifically, the C_C cluster, which encompasses Germany, Estonia, Greece, Latvia, Lithuania, Hungary, Austria, Slovenia, and Finland, holds the second position in this indicator with a score of 24.97%. This place is confirmed by the third place occupied by the indicator at-risk-of-poverty rate PPR (15.51%), with which it is inversely proportional (validation I1), and by the level of internet access households with 91.32% (validation I2). In terms of educational attainment, the substantial percentages of 49.73% for PPEL34 (second place) and 32.91% for PPEL58 (third place) indicate a high level of the population with primary, secondary and tertiary education (PECO), and the 17.39% in the fourth place occupied by PPEL02 does not have a great influence on the PECO, as this indicator is not very important considering the reduced adaptability to e-commerce activities.

In the aftermath of the COVID-19 pandemic, these nations implemented strategic economic measures aimed at fostering economic recovery through the utilization of e-commerce. However, the outcomes pertaining to the advancement of e-commerce did not align with initial expectations. The population adapted to the new requirements related to e-commerce, but not at such a fast pace as it happened in the cluster A countries.

Spain, Italy, Malta, and Portugal form the C_D cluster, which, with 23.15% of the PECO, is placed in the middle of the cluster ranking (third place). The four nations have encountered significant economic challenges stemming from the COVID-19 pandemic. Despite concerted efforts, their efforts to stimulate recovery through an emphasis on e-commerce activities have been largely unsuccessful. This assertion is substantiated by the alterations made to the variables implemented in the study. In 2021, following the period of the global pandemic, it was observed that the countries faced a considerable poverty-risk rate of 16.68% and a household internet access level of 91.07%, which placed them fourth

globally. This situation poses significant challenges for economic activities within PECO. The same situation for PECO is indicated by the very low level (last place) of PPEL34 with 34.13% and the reduction by the 27.98% of PPEL58 (fourth place). The first place occupied in the ranking of the media clusters of these countries regarding PPEL02 (37.85%) does not have a great influence on PECO.

Table 6 shows the characteristics of clusters C_B and C_E, which occupy the last two places in terms of the PECO variable (19.15% cluster C_B, and 16.43% cluster C_D).

The two clusters include the countries with the lowest level of e-commerce after the end of the COVID-19 pandemic. Similar to the preceding examination of the three clusters, the present discussion endeavors to underscore the extent of the additional variables and their interconnection with PECO.

Six other countries (Bulgaria, Czechia, Croatia, Poland, Romania, Slovakia) form the C_B cluster, which corresponds to a reduced PECO of only 19.05%, which places it fourth in the cluster hierarchy. The observed outcome is attributed to the impact of modifications implemented in response to the COVID-19 pandemic, as recorded in the year 2021. With a relatively limited level of economic development, particularly in the aftermath of the pandemic, there is a high risk of poverty (15.78%, ranking second highest) and a relatively low level of household internet access (88.34%, ranking lowest), an unfavorable situation for the activities carried out by PECO. The peculiarity of this cluster lies in the particular situation that occurs with regard to PPEL34. The cluster PPEL34 holds the highest weight of 60.20% in the ranking; however, this indicator does not make a substantial contribution to the growth of PECO. The rationale primarily pertains to the limited degree of economic advancement observed in the respective nations during the specified year. This is particularly noteworthy given the comparatively low emphasis placed on the remaining two tiers of education, highlighting their position at the bottom of the hierarchical structure.

The cluster denoted as C_E exhibits the lowest percentage of 16.43%, specifically attributed to e-commerce activities, customer relationship management (CRM), and secure transactions. This positions the C_E cluster at the bottom of the hierarchical ranking among the five clusters. This means that the countries that have suffered the most from the COVID-19 pandemic, without quickly recovering from an economic point of view through PECO activities, are France, Cyprus, and Luxembourg. This cluster is characterized by two peculiarities in the sense that, although the weight of the at-risk-of-poverty rate is the lowest (10.07%), in the case of these countries, this indicator does not have a significant impact on PECO. The same aspect can be highlighted in the case of the PPEL indicator, both for PPEL58 with 40.90% (first place) and through PPEL02 with 22.30% (second place), which should have contributed significantly to the increase in the share of PECO. However, the economic situation and the factors of economic growth in the context of the exit from the pandemic have a much stronger impact on PECO than the indicators specified for these countries. This has resulted in a detrimental impact on the activities within the PECO. Considering their significant impact, the percentage values of the other two variables also demonstrated a positive association with PECO, as evidenced by the lower mean values of level of internet access—households (95.31%, second place) and PPEL34 (36.80%, fourth place).

The findings of this section demonstrate that the examination of the indicators conducted by country and cluster reveals that similarities in the direction of the indicators are reflected by closely aligned values, while divergent trends are reflected by significant disparities between their means. Based on the obtained results, it can be inferred that developed nations exhibit a notable level of e-commerce utilization, thereby supporting hypothesis H1. The extent of the variation in the average indicators emphasizes the support for hypothesis H2, suggesting that countries with internet access utilize it in accordance with the educational level of their population. This assertion is corroborated and augmented by the results of indicators pertaining to the proportions of the population across three education categories: PPEL02, PPEL34, and PPEL58. Concurrently, the findings regarding the mean values of the indicators underscore the association between a height-

ened standard of living and a diminished risk of poverty, which in turn correlates with an increased utilization of electronic commerce, as posited in hypothesis H3.

## 5. Discussion

Considering the specified variables, the subsequent discussion provides a clearer synthesis of the findings, with a focus on the average percentage values across the analyzed clusters and variables. Therefore, their interpretation elucidates the similarities and differences that are evident among the clusters.

In alignment with prior investigations [62], the findings demonstrate the presence of a subset of European Union nations exhibiting a robust e-commerce infrastructure, comprising Belgium, Denmark, Ireland, the Netherlands, and Sweden, in juxtaposition to another cohort of nations with a less advanced e-commerce infrastructure, including Bulgaria, the Czech Republic, Croatia, Poland, Romania, and Slovakia (see Table 7).

**Table 7.** Synthesis of the cluster analysis discussion.

| | | |
|---|---|---|
| C_A | Belgium | The pinnacle of electronic commerce, a minimal level of poverty risk, the highest degree of household internet access, a percentage below the European norm for individuals with limited educational attainment and those with secondary education, coupled with a significant proportion of individuals with tertiary education. |
| | Denmark | |
| | Ireland | |
| | The Netherlands | |
| | Sweden | |
| C_B | Bulgaria | The level of e-commerce in this region falls below the European average, accompanied by a heightened risk of poverty. Moreover, it exhibits the lowest level of household access to the internet, the lowest percentage of the population with lower education, and the highest percentage with secondary education, as well as the lowest share of the population with higher education. |
| | Czechia | |
| | Croatia | |
| | Poland | |
| | Romania | |
| | Slovakia | |
| C_C | Germany | A high level of e-commerce, a level of poverty risk above the European average, a level of household access to the internet below the European average, a low share of the population with lower education, a high share of the population with secondary education and a share of the population with higher education above the European average. |
| | Estonia | |
| | Greece | |
| | Latvia | |
| | Lithuania | |
| | Hungary | |
| | Austria | |
| | Slovenia | |
| | Finland | |
| C_D | Spain | A level of e-commerce above the European average, the highest level of poverty risk, a low level of household access to the internet, the highest share of the population with lower education, the lowest share of the population with secondary education, as well as a small share of the population with higher education |
| | Italy | |
| | Malta | |
| | Portugal | |
| C_E | France | The lowest level of e-commerce, the lowest level of poverty risk, a high level of household access to the internet, a share of the population with lower education below the European average, a low share of the population with secondary education, as well as the largest proportion of the population with higher education. |
| | Cyprus | |
| | Luxembourg | |

Source: elaborated by authors.

E-commerce might be preferred by the poor population because, on the internet, the products are at lower prices; however, they do not adopt it, possibly due to the fact that they do not have access to the internet and do not have the necessary skills. According to some studies, the lack of access to the internet can be considered a barrier due to its associated cost, particularly for those with lower socioeconomic status [63]. The phenomenon of poverty exerts a direct impact on e-commerce by virtue of its effect on purchasing power, and indirectly through the limitations it imposes on individuals' ability to invest in their own personal development, including the acquisition of internet-related skills [64]. There are even scholarly investigations that utilize e-commerce data to gauge levels of poverty [65].

E-commerce is significantly impacted by the availability of infrastructure, as economic agents are required to have access to the internet in order to conduct online transactions. This assertion is further supported by other scholarly literature [66], which posits that an improved infrastructure and higher skill level among the populace will lead to an uptick in the utilization of e-commerce.

A study underscored the significance of internet access, particularly broadband access, in fostering economic development [67].

In order to initiate a purchase, individuals must possess proficiency in operating a personal computer or mobile device, which may necessitate fluency in foreign languages, particularly English, and an understanding of internet navigation. As educational attainment increases, individuals are more inclined to utilize the internet and engage in online transactions. This is further substantiated by other existing research studies [68,69].

Even a study carried out by Eurostat highlights that there is positive correlation with education level and employment status. It says that education level seems to be positively correlated with e-commerce. In 2022, 56% of people with a low level of formal education bought online, but this rate increased to 74% for people with medium level of formal education and even 88% for people with higher education [70].

Cluster C_A comprises of the most economically advanced nations in terms of the GDP per capita, all of which surpass the European average, with Ireland ranking second after Luxembourg. Although the purpose of the study is not to highlight the links between GDP and e-commerce, it can still be noted that there is a link between the level of development of a state and the level of e-commerce (Parishev, etc.). In consideration of the findings pertaining to the C_B cluster variables, the analysis accounts for the observation that the variable denoting the highest level of education (PHEL 58) carries the least weight, thereby being ranked in the lowest position within the hierarchy. It is apparent that the cluster under consideration comprises solely nations with a history of socialism, many of which exhibit a relatively low level of development, as evidenced by their low GDP per capita. The countries constituting the C_B cluster (Bulgaria, Czech Republic, Croatia, Poland, Romania, Slovakia) exhibit a relatively low prevalence of businesses engaging in electronic commerce, with the average adoption rate of this cluster falling below the European average. This observation may underscore the persistent disparities between these nations and capitalist economies, despite the implementation of diverse economic restructuring efforts. It should be noted that two European states, the Czech Republic and Croatia, exceed the average e-commerce levels in the region, indicating that they are making progress and may soon close the gap. It is notable that within this cluster, there are two countries with significantly poor performance, with Bulgaria ranking last and Romania ranking second to last.

The C_C cluster exhibits significant heterogeneity, comprising both developed countries such as Germany, Finland, and Austria, as well as former socialist countries, notably the Baltic states. Estonia, in particular, has garnered attention for its successful implementation of rigorous reforms, serving as a model for other former socialist states to emulate. It is plausible that the early accession to the European Union, resulting in increased financial support, and the adoption of the euro currency, facilitating transactions with other European countries, may have contributed to this achievement.

The findings provide additional evidence to support the hypothesis of a causal relationship between the levels of economic development and e-commerce activities [71–74].

Although this C_E cluster seems to disprove the rule that there is a direct link between developed countries and e-commerce, we believe that this cluster presents some peculiarities rather than a deviation (exception) from the rule.

The low prevalence of e-commerce in Luxembourg, which ranks highest in GDP per capita within the EU, may be attributed to the exclusion of data from the financial sector in Eurostat's data collection. This omission accounts for approximately 25% of Luxembourg's GDP and is not reflected in the e-commerce activity of the companies under analysis. Moreover, the steel industry in Luxembourg is highly developed, and it is not well-suited for online commerce.

In the case of Cyprus, permissive legislation most likely encourages the offshore industry. It is tax haven—companies only register there; they have no economic activity and by default no e-commerce. In the French context, it is feasible that the considerable presence of influential state-owned enterprises within the economy has exerted an influence on the proportion of online sales.

Similar to the previously published literature on this topic, this study highlighted the negative influence of poverty on e-commerce [63,65], the fact that infrastructure has a positive effect on e-commerce [66,67], as well as the fact that the degree of higher education increases the chances of conducting electronic commerce [68–70].

Additionally, the outcomes of our investigation align with the findings of other scholarly works, indicating the presence of a subset of European Union nations possessing a robust e-commerce framework, such as Belgium, Denmark, Ireland, the Netherlands, and Sweden, in contrast to a separate faction of countries with a less advanced e-commerce landscape—countries encompassing a trade infrastructure, including Bulgaria, Czech Republic, Croatia, Poland, Romania, and Slovakia [62].

## 6. Conclusions

The strategy of forced digitization manifested after 2020 can be attributed to the profound influence of the COVID-19 pandemic on the economic transformations experienced globally, particularly within European nations. The advancement of electronic commerce, as an integral aspect of the digitalization approach, is imperative for the economies of contemporary nations. Hence, it is imperative to conduct an examination of its representation at both the individual level within each country, contingent upon the level of development and perceptions of the ramifications of the pandemic, and at a global level by means of comparative analysis among European Union countries. This will allow for the formulation of a comprehensive understanding regarding the extent of electronic commerce development.

In this context, the issue of highlighting the similarities and disparities in e-commerce among the 27 member states of the European Union, subsequent to the COVID-19 pandemic, was explored in the year 2021. The study involved an analysis of six variables, which were examined using a two-way design consisting of three levels for each variable. The initial proposal encompassed components related to economic entities involved in e-commerce activities, the vulnerability to poverty, and the educational attainment of the populace (specifically focused on e-commerce, customer relationship management (CRM), secure transactions, at-risk-of-poverty ratio, and household-level internet accessibility). The second scheme illustrated the educational attainment of the overall population according to three distinct categories of education: individuals with education less than primary level, those with primary and lower secondary education (levels 0–2), those with upper secondary and post-secondary non-tertiary education (levels 3 and 4), and those with tertiary education (levels 5–8).

The hierarchical cluster methodology was employed to analyze the variables, incorporating the use of the Ward method alongside the dendrogram output. To ascertain the statistical significance of the means derived for the five clusters formed, Robust Tests of Equality of Means (Welch and Brown–Forsythe) were conducted.

The analysis revealed significant findings regarding the prevailing or ultimate positions of the EU clusters and countries within the hierarchies generated through the

implementation of the clustering method. These findings elucidate the impact of indicators on electronic commerce, emphasizing both advantageous and detrimental effects. In the field of electronic commerce, certain factors, such as customer relationship management (CRM) and secure transactions, have emerged as critical components. The C_A cluster, comprised of Belgium, Denmark, Ireland, Netherlands, and Sweden, stands out as having the highest level of adaptation to meeting market requirements, thereby contributing to a notable progression in the domain of electronic commerce. The high level is further corroborated by the indicator of household internet access. Ireland emerges as the foremost nation in terms of vigorous e-commerce participation, not only among the listed countries but also among all the other European Union (EU) nations. Conversely, Luxembourg is distinguished as the European country with the most extensive internet connectivity, specifically in terms of household access. It is important to acknowledge that the variable concerning the at-risk-of-poverty rate exhibits the highest weights, which exert a detrimental influence on e-commerce activity. Consequently, countries with the lowest weights possess the ability to significantly impact e-commerce activity. The C_E cluster, composed of France, Cyprus, and Luxembourg, exhibits the lowest average weight for the at-risk-of-poverty rate. Nonetheless, within the European Union, the Czech Republic shows the lowest weight in this regard, consequently exerting the most positive influence on e-commerce activity based on this indicator.

The analysis of the cluster and country rankings based on the proportion of the total population possessing different levels of education post-COVID-19 in 2021 holds significant importance. Particularly, nations with a substantial percentage of individuals attaining tertiary education and upper secondary and post-secondary non-tertiary education (levels 3 and 4) exert a notable influence on e-commerce activity. Conversely, the demographic comprising individuals with education levels below primary, primary, and lower secondary (levels 0–2) emerges as a potential force driving the advancement of electronic commerce. Therefore, it can be observed that France, Cyprus, and Luxembourg, categorized as cluster C_D, exhibit the most notable mean proportion of their population possessing a tertiary level education. However, on a broader scale within the European Union, Ireland emerges as particularly noteworthy in this regard. The C_B cluster, comprising Bulgaria, Czech Republic, Croatia, Poland, Romania, and Slovakia, exhibits the largest proportion of individuals possessing upper secondary and post-secondary non-tertiary education (specifically levels 3 and 4). Among these countries, the Czech Republic particularly distinguishes itself in this regard, within the context of the European Union. In the aftermath of the COVID-19 pandemic, the countries belonging to the C_D cluster, namely Spain, Italy, Malta, and Portugal, have emerged as noteworthy players in shaping the future of e-commerce. Notably, Portugal has assumed a dominant position at the European level in this regard.

These positions explain the gradual and balanced tendency to adapt to the digitization process, coupled with a willingness to formulate and implement e-commerce growth strategies in support of expediting economic revival within nations.

The analysis conducted at the European Union level revealed that the outcomes obtained necessitate the implementation of measures designed to expedite and align with the demands of the digitized product market in countries where electronic commerce exhibits considerable fragility. These initiatives are predominantly derived from the utilization of European funds, necessitating member states to allocate resources toward investment in the following:

➢ The communications infrastructure, increasing the high-speed internet coverage areas.
➢ Education, by creating the necessary infrastructure for free access to the internet in all educational units. Also, it is proposed that the early inclusion of computer science and foreign language instruction within the school curriculum is imperative, as it is essential for facilitating broad and unrestricted utilization of the internet.
➢ One potential means of providing support to individuals facing vulnerability is by offering vouchers that can be utilized to procure devices enabling access to the internet.

Another practical consideration derived from this study pertains to the strategies for boosting online sales. The expansion of e-commerce can be significantly enhanced through the development of digital proficiency and increased assurance in online security measures. The findings of a survey conducted among the adult population of Romania indicate that approximately 54% of respondents refrain from making online purchases. The primary reasons cited for this behavior include a lack of trust in online payment methods, reported by 33% of respondents, and a perceived lack of necessary skills, noted by 21% of respondents [75]. It is evident that countries prioritizing digitization and electronic security have achieved advanced levels of electronic commerce, providing empirical support for this hypothesis. In order to enhance their alignment with market demands, companies ought to prioritize the allocation of resources toward human capital development [76]. The provision of training courses/programs for employees is imperative to enhance their computer proficiency, facilitating a smoother transition to the online working environment. Organizations must also allocate resources toward the development of information technology infrastructure essential for facilitating electronic commerce and communication, thereby enhancing customer convenience and operational effectiveness [77].

*Limitations and Further Research*

The examination of the similarities and disparities among the clusters and nations provided a comprehensive analysis of the progression of e-commerce based on the thoroughly investigated variables. Nevertheless, certain limitations exist, with the purpose of conducting a more concise, intricate, and comprehensive assessment of the extent of electronic commerce's progression. The primary factor relates to the restricted availability of the database, particularly exacerbated against the backdrop of the COVID-19 pandemic. Consequently, the task of selecting variables becomes arduous and, at times, even unattainable due to their paucity. This, in turn, hinders the possibility of adequately examining the interrelationships among specific factors that exert considerable influence on e-commerce activity. One potential constraint may pertain to the duration of the analysis, which solely encompassed the calendar year of 2021, deemed the most pivotal period following the conclusion of the COVID-19 pandemic. Simultaneously, this study may be conducted on a global scale; however, it is presently limited to European countries.

It is important to acknowledge that the findings of this research are subject to certain constraints, such as sample size and data collection constraints. These limitations should be carefully considered when interpreting the results and drawing conclusions. Further research in this area is warranted to address these limitations and expand upon the findings of this study. Future studies could explore alternative methodologies, larger sample sizes, and different data collection techniques to build upon the current research and provide a more comprehensive understanding of the topic. Additionally, investigating any potential biases and addressing them in future research is essential to ensure the validity and reliability of the findings.

The present study's analysis is limited to a single year and recommends the extension of the time frame to encompass multiple years, both preceding and following the COVID-19 pandemic. Such an extension would allow for a more comprehensive understanding of the dynamics of various states, facilitating an examination of their performance and pace of development in the realm of electronic commerce.

This study is limited in scope due to the fact that the analysis was conducted solely for EU countries, thus failing to capture the varying degrees of e-commerce development globally. This omission precludes the identification and comparison of e-commerce development in different regions and among countries with differing levels of e-commerce advancement, potentially even at a continental scale.

One overarching constraint of this research is the restricted utilization of variables in the analysis, indicating the potential for broader exploration of additional factors influencing the augmentation of value, volume, and efficiency within the realm of electronic commerce.

**Author Contributions:** Conceptualization, R.M.G., L.C.M., D.A.B. and L.M.P. and C.F.V.; methodology, R.M.G.; software, R.M.G.; validation, R.M.G.; formal analysis, R.M.G.; investigation, R.M.G. and L.C.M.; resources, R.M.G. and L.C.M.; data curation, R.M.G.; writing—original draft preparation, R.M.G. and L.C.M.; writing—review and editing, R.M.G., L.C.M. and C.F.V.; visualization, D.A.B., L.M.P. and C.F.V.; supervision, D.A.B., L.M.P. and C.F.V.; project administration, R.M.G. and L.C.M. All authors have read and agreed to the published version of the manuscript.

**Funding:** This research received no external funding.

**Institutional Review Board Statement:** Not applicable.

**Informed Consent Statement:** Not applicable.

**Data Availability Statement:** The raw data are available from the corresponding author upon reasonable request.

**Conflicts of Interest:** The authors declare no conflicts of interest.

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
