# Peer review of "Similarities and Disparities of e-Commerce in the European Union in the Post-Pandemic Period"

_jtaer, doi:10.3390/jtaer19010018_

Round 1

Reviewer 1 Report (Previous Reviewer 1)

Comments and Suggestions for Authors

The authors have diligently addressed the concerns raised in the previous review process, demonstrating a thorough commitment to refining their work. As a result, the paper is now deemed ready for publication in its current state

Author Response

We truly appreciate your feedback as it helps us improve our article and we are grateful for your kind words of encouragement.

Best wishes to you and please don't hesitate to reach out if you have any further suggestions or inquiries.

Reviewer 2 Report (Previous Reviewer 2)

Comments and Suggestions for Authors

Dear Authors,

I received your article "Similarities and disparities of e-commerce in the European Union in the post pandemic period" sent to Journal of Theoretical and Applied Electronic Commerce Research and I have the following improvement recommendations.

1. At the lines 86 - 87 you have the text: "For this statistical-econometric research, the accuracy of the results obtained is given by the confidence level of 95%, used to verify (test) the research hypotheses:", then you have an enumeration of three proposed research hypotheses. I recommend you the followings:

- rename the hypotheses to H1, H2, H3 because the scientific literature uses these type of notations instead of I1, I2, I3.

- the research hypotheses should be moved from the Introduction section to the end of the Literature Review section, because they should be based on the previous results from the literature.

2. In the Introduction you should describe the research gap. I have read and re-read this section, but the research gap is unclear. Please add a new paragraph to clearly define the research gap. The research gap should be based on some previous resources from the literature.

3. At the lines 323 - 325, in the section "3. Data series and Methodology" you have the text: "In order to identify and highlight the disparities and similarities between European states regarding the impact of education for poverty on the level of e-commerce, the research methodology included the following stages:". Then, you have only one stage at the line 326: "Identification and collection of variables included in the research". Within this stage, you have many sub-stages with bullets. Please revise this issue, because you talk about "stages", but in fact you describe only one main stage with different sub-stages. I suggest you to re-organize this sequence.

4. In Table 1 you present the list of variables included in the analysis. Please present under the table some arguments for this choices. Why these variables and not other?

5. In the sequence from the lines 551 - 563, please replace I1, I2, I3 with H1, H2, H3 (see the first recommendation from this review report).

6. There is a blank space between the lines 600 - 601. Please revise and remove it.

Best regards!

Author Response

This manuscript is a resubmission of an earlier submission. The following is a list of the peer review reports and author responses from that submission.

Round 1

Reviewer 1 Report

Comments and Suggestions for Authors

I have reviewed the paper titled "Similarities and Disparities of E-commerce in the European Union in the Post-pandemic Period," and I would like to provide my feedback and observations on the paper. While the paper addresses an important topic, there are several areas that need improvement to enhance its quality and academic rigour.

1.      The introduction section of the paper fails to adequately justify the rationale for conducting the research. It is crucial to provide a clear and compelling argument for why this study is needed. The authors should clarify the research gap that their work aims to fill and explain the significance of their research in the context of post-pandemic e-commerce in the European Union.

2.      The literature review is extensive, but it lacks a clear identification of the gaps in existing research. While it is important to provide a comprehensive background, it is equally important to highlight specific areas where previous studies fall short and how the current study intends to address those limitations.

3.      The paper does not specify the data period under consideration. It is essential to provide a clear timeframe for the data collection and analysis, especially when discussing post-pandemic trends, as the dynamics of e-commerce may change over time.

4.      How can a single hypothesis be tested based on multiple p-values (H0)

5.      The discussion section primarily summarizes the findings of the current study, but it fails to integrate and critically analyze the existing literature. The authors should discuss how their findings relate to previous research, highlight differences or similarities, and provide insights based on the broader research landscape.

6.      Lacks a discussion of the implications of the study. It is crucial to briefly summarize the key findings and then elaborate on the practical and theoretical implications of the research.  

            Comments on the Quality of English Language

No

Reviewer 2 Report

Comments and Suggestions for Authors

Dear Authors,

Your article "Similarities and disparities of e-commerce in the European Union in the post pandemic period" can be improved according to the following recommendations.

1. The Introduction section should be improved because it needs clear descriptions for the following elements in a scientific article:

- the research gap (at this moment, the readers don't understand the gap covered by your proposal);

- the research goal (please clearly define the general goal of the article, according to the research gap);

- the research questions (you should define and describe the research questions derived from the research goal).

Please add distinct paragraphs for the above mentioned aspects, so that the readers have a clear image of your proposal.

2. The section "2. Literature review" is too long and it doesn't focus on your theme. It contains too much "history of e-commerce".

I recommend you to remove some irelevant paragraphs and to focus on the following aspects:

- at the end of the Literature Review section, define and describe the research hypotheses. Now, the research hypotheses are very diffuse.

- include the following relevant resources in your work: https://doi.org/10.3390/businesses2020017 (marketing and technology), https://doi.org/10.3390/electronics12132857 (Influence of Network Centrality and Density), https://doi.org/10.3390/su12176993 (e-commerce in Europe), https://doi.org/10.3390/economies9020057 (e-commerce and pandemic). These new resources will really improve the background of your manuscript.

3. In the section "3. Data series and Methodology" you have "Table 1. The list of variables, by waste category, included in the analysis". The title refers to "... waste category ...". Waste means trash and I don't understand this notation. Please revise and correct it.

4. Please explain the methodology of choosing the variables from the table 1, because at this moment it seems that these variables are based on your own intuition. Please provide some sources from the literature.

5. I recommend you to clearly describe the methodology (the steps) at the beginning of the section 3.

6. At the end of the Results section, you should provide information about the research hypotheses. Please explain if the research hypotheses are supported or not supported.

7. In the Discussion section, please compare your research results to the other from the literature.

8. The lines 791 - 792 are empty. Please remove them.

Best wishes!

Round 2

Reviewer 2 Report

Comments and Suggestions for Authors

Dear Authors,

I appreciate your efforts to improve the article.

However, some minor aspects should be addressed at this moment:

1. In the Conclusions section, please include a distinct paragraph describing the limitations of your research.

2. English should be revised in the case of some sentences.

Best regards!

Author Response

Dear Reviewer 2,

Here are our answers to the aspects you highlighted: 

  1. In the Conclusions section, please include a distinct paragraph describing the limitations of your research.

Several aspects pertaining to the constraints of the study were not adequately emphasized, however, following your recommendation, we have reorganized and articulated them in the Conclusions section. We are grateful for your input in this matter.

Limitations and Further Research

It is important to acknowledge that the findings of this research are subject to certain constraints, such as sample size and data collection constraints. These limitations should be carefully considered when interpreting the results and drawing conclusions. Further research in this area is warranted to address these limitations and expand upon the findings of this study. Future studies could explore alternative methodologies, larger sample sizes, and different data collection techniques to build upon the current research and provide a more comprehensive understanding of the topic. Additionally, investigating any potential biases and addressing them in future research is essential to ensure the validity and reliability of the findings.

The present study's analysis is limited to a single year and recommends the extension of the timeframe to encompass multiple years, both preceding and following the Covid-19 pandemic. Such an extension would allow for a more comprehensive understanding of the dynamics of various states, facilitating an examination of their performance and pace of development in the realm of electronic commerce.

This study is limited in scope due to the fact that the analysis was conducted solely for EU countries, thus failing to capture the varying degrees of e-commerce development globally. This omission precludes the identification and comparison of e-commerce development in different regions and among countries with differing levels of e-commerce advancement, potentially even at a continental scale.

One overarching constraint of this research is the restricted utilization of variables in the analysis, indicating potential for broader exploration of additional factors influencing the augmentation of value, volume, and efficiency within the realm of electronic commerce.

  1. English should be revised in the case of some sentences.

We have revised the English technical drafting, but if we have missed something, please let us know the sentences that would require further revisions.

We express our gratitude for the recommendations received and have strived to incorporate them in a manner that enhances the quality of the article in line with the publication's standards.

To this message we also attached the revised version of the paper.

Thank you for your time.

Best wishes